# Evolution of gene expression levels in the male reproductive organs of *Anopheles* mosquitoes

Abril Izquierdo[1,*], Martin Fahrenberger[1,*], Tania Persampieri[2], Mark Q Benedict[3], Tom Giles[4], Flaminia Catteruccia[5], Richard D Emes[1,4] , Tania Dottorini[1]

**Modifications in gene expression determine many of the phenotypic differentiations between closely related species. This is particularly evident in reproductive tissues, where evolution of genes is more rapid, facilitating the appearance of distinct reproductive characteristics which may lead to species isolation and phenotypic variation. Large-scale, comparative analyses of transcript expression levels have been limited until recently by lack of inter-species data mining solutions. Here, by combining expression normalisation across lineages, multivariate statistical analysis, evolutionary rate, and protein–protein interaction analysis, we investigate ortholog transcripts in the male accessory glands and testes across five closely related species in the *Anopheles gambiae* complex. We first demonstrate that the differentiation by transcript expression is consistent with the known *Anopheles* phylogeny. Then, through clustering, we discover groups of transcripts with tissue-dependent expression patterns conserved across lineages, or lineage-dependent patterns conserved across tissues. The strongest associations with reproductive function, transcriptional regulatory networks, protein–protein subnetworks, and evolutionary rate are found for the groups of transcripts featuring large expression differences in lineage or tissue-conserved patterns.**

## Introduction

Gene expression levels are quantitative molecular traits that link genotype to phenotype through modulation of molecular functions, regulating and affecting cellular and organismal fitness. Changes in gene expression drive and are driven by phenotypic differentiation and characterise divergence across species (Rifkin et al, 2003; Blekhman et al, 2008; Brawand et al, 2011; Xu et al, 2018). However,

our understanding of how genome evolution is related to expression and phenotype modification is still predominantly qualitative. For example, we do not know whether there is a threshold above which expression elicits a functional effect, if there are biases towards the modulation of different functional classes, and how expression changes propagate through the interactome, ultimately influencing the tissue, the individual, and the entire species.

The investigation of gene expression differences in rapidly evolving genes amongst closely related species allows dissection of evolution of the genome, expression, and phenotypic differentiation. In particular, comparative studies focusing on gene expression differences in reproductive tissues of closely related species, where genes evolve rapidly (Swanson & Vacquier, 2002; Nielsen et al, 2005; Clark et al, 2006; Haerty & Singh, 2006; Lemos et al, 2007; Moehring et al, 2007; Turner & Hoekstra, 2008; Sundararajan & Civetta, 2011), provide insights into how genome evolution influences gene expression changes at the tissue, sex, and species-specific level and how such changes are functionally, anatomically, and physiologically integrated into and linked to fitness.

Understanding changes in gene expression of reproductive tissues of divergent *Anopheles* mosquitoes provides insight into the phenotypes that drive differences in these important disease vectors. Females of most anopheline species generally mate only once during their lifetime. In *Anopheles gambiae*, as in many other insects, the male accessory glands (MAGs) produce seminal secretions, including a class of MAGs-specific proteins known as Acps (accessory gland proteins) and the steroid hormone 20-hydroxyecdysone (20E), that once transferred to the female during copulation coagulate into a mating plug (Dottorini et al, 2007; Rogers et al, 2009; Mitchell et al, 2015). The transfer of the plug and its content has profound effects on female physiology as it induces a cascade of events that render *An. gambiae* females refractory to further copulation, enhance egg production, and trigger egg laying (Thailayil et al, 2011; Dottorini et al, 2012; Gabrieli et al, 2014; Mitchell et al, 2015). The catalytic activity of the transglutaminase enzyme TG3 AGAP09099 is responsible for mating plug coagulation through the protein plugin

[1]School of Veterinary Medicine and Science, Sutton Bonington Campus, University of Nottingham, Leicestershire, UK  [2]Department of Experimental Medicine, University of Perugia, Perugia, Italy  [3]Centers for Disease Control and Prevention, Division of Parasitic Diseases and Malaria, Entomology Branch, Atlanta, GA, USA  [4]Advanced Data Analysis Centre, Sutton Bonington Campus, University of Nottingham, Leicestershire, UK  [5]Department of Immunology and Infectious Diseases, Harvard T. H. Chan School of Public Health, Boston, MA, USA

Correspondence: tania.dottorini@nottingham.ac.uk
*Abril Izquierdo and Martin Fahrenberger contributed equally to this work.

AGAP009368 (Acp). Such catalytic activity has been found to vary across different anopheline species (Neafsey et al, 2015). In recent work (Mitchell et al, 2015), correlation between 20E abundance (and thus, indirectly, the expression of genes in the 20E synthesis pathway) and the attributes of shape and consistency of the mating plug has been demonstrated between species, with the plug varying from solid and highly coagulated, to less compact and amorphous to being absent in some species. These findings hint that observed phenotype variations of reproductive strategy can be linked to expression of specific genes, across species, underlying the involved molecular pathways.

To date, our understanding of the relationship of transcription between genes encoding proteins involved in *Anopheles* reproduction and different phenotypic manifestations, across development, tissues, sexes, individuals, and species is limited, and hence, the studies correlating variation of gene expression to phenotype modification are of fundamental importance. In Dissanayake et al (2006), developmental gene expression (larvae, adult) of the single *An. gambiae* species was investigated between sexes, as was expression in fat body, midgut, and ovaries of females before and after a blood meal. Previously, we investigated gene expression of sex-biased genes of *An. gambiae* in relation to developmental stages (Magnusson et al, 2011), again focusing on a single species. Recent studies have included comparative analyses across multiple anopheline species: the 16 Genomes Project (Neafsey et al, 2015) and 8 in Mitchell et al (2015). A recent multispecies comparison (Papa et al, 2017) investigated the expression differences between sexes of four species (*An. gambiae, An. arabiensis, An. minimus*, and *An. albimanus*) and considered nonreproductive tissue (carcasses) and combined reproductive tissue (MAGs and testes in males, and ovaries and common oviduct in females). Because of this aggregation, it was not possible in such work to obtain information specific to the individual reproductive tissues.

To address the lack of resolution of individual reproductive tissues, we have focused only on males and explored MAGs and testes gene expression separately in five closely related Anopheline species from the *An. gambiae* sensu lato complex. The correct phylogeny of this complex has remained contentious until recently (Fontaine et al, 2015). This sub-Saharan complex of several closely related species, includes the world most important vectors of human malaria in *An. gambiae* s.s., *An. coluzzii*, and *An. Arabiensis*, as well as minor or locally important vectors (*An. merus*) and nonvectors (*An. quadriannulatus*). The three anthropophilic species *An. gambiae* s.s., *An. coluzzii*, and *An. arabiensis* are sympatric and spread in a vast Afrotropical area. *An. merus* lives in coastal waters of eastern Africa, whilst *An. quadriannulatus* lives in Southern Africa and bites animals other than humans. Within the entire *Anopheles* genus, the amount of important malaria vectors is relatively small. However, almost all of them are embedded in complexes similar to those of *An. gambiae* sensu lato (Manguin, 2013). Thus, the investigation of the *An. gambiae* complex can possibly provide insight into the biology of all the malaria vectors across the entire genus.

The choice of focusing on closely related species within the same complex is driven by gene expression changes evolving rapidly compared with DNA sequence variations (Pollard et al, 2006; Xu et al, 2018). Thus, a comparison of distant lineages could be dominated by significant DNA differences leading to imprecise

assignment of species and transcriptional changes. Most importantly, differently from previous work, for example, Papa et al (2017), we have adopted a method for normalising transcript expressions across species which, combined with multivariate statistical analysis, allows a comparative investigation of expression levels not only between tissues but also across different species. In this work, the observation of how expression changes within the *An. gambiae* complex is combined with prediction of molecular function, study of the transcriptional regulatory networks, and other protein–protein interactions, and to the study of evolutionary rates.

# Results

## Generation of the male reproductive transcriptome and comparison across species

To study the evolutionary dynamics of gene expression in male reproductive tissues across the *An. gambiae* species complex, we generated transcriptomes of the MAGs and testes from five different anopheline mosquitoes: *An. gambiae* s.s., *An. coluzzii*, *An. arabiensis*, *An. merus*, and *An. quadriannulatus*. In each reproductive tissue, we measured RNA expression in 30 mosquitoes. Sequencing resulted in annotation of approximately 13,000 transcripts in each species (Table S1). The majority of exons and introns are known, but a substantial improvement of the coverage and transcriptome assemblies was obtained (Table S1).

## Inter-species expression normalisation and classification of transcripts on expression levels

To allow comparison across species, the expression levels were normalised using a method originally described in Brawand et al (2011), based on computing weighting factors. Weighting factors were obtained by identifying a subset of ortholog transcripts whose expression levels remain the most constant across datasets. These selected orthologs were used to form a reference expression level against which all the other genes were scaled (see the Materials and Methods section). Analysis of the relative magnitude of expression of transcripts in each tissue, allowed transcripts to be classified as ubiquitous, enriched, or highly enriched within MAGs and testis (Fig 1A) for each species. Classification used relative expression in a tissue computed as $\log_2$ (fold change +1) (referred to as $\log_2$FC in the following): highly enriched genes were those with $\log_2$FC ≥ 3.5; enriched genes were those with $2 \leq \log_2$FC < 3.5, and ubiquitous genes were those with $-2 \leq \log_2$FC < 2. Apart from *An. coluzzii*, the proportion of enriched or highly enriched genes is greater in the testes than in the MAGs (Fig 1A). The MAG-specific gene expression is also different, with a high proportion of very lowly expressed genes coupled with a few often very highly expressed genes (Fig 1B).

## Summary of the main results

The analysis involved 5,493 one-to-one (1:1) ortholog transcripts encompassing the five species and the two selected tissues (MAGs and testes). These transcripts represent 38.50% of the total found

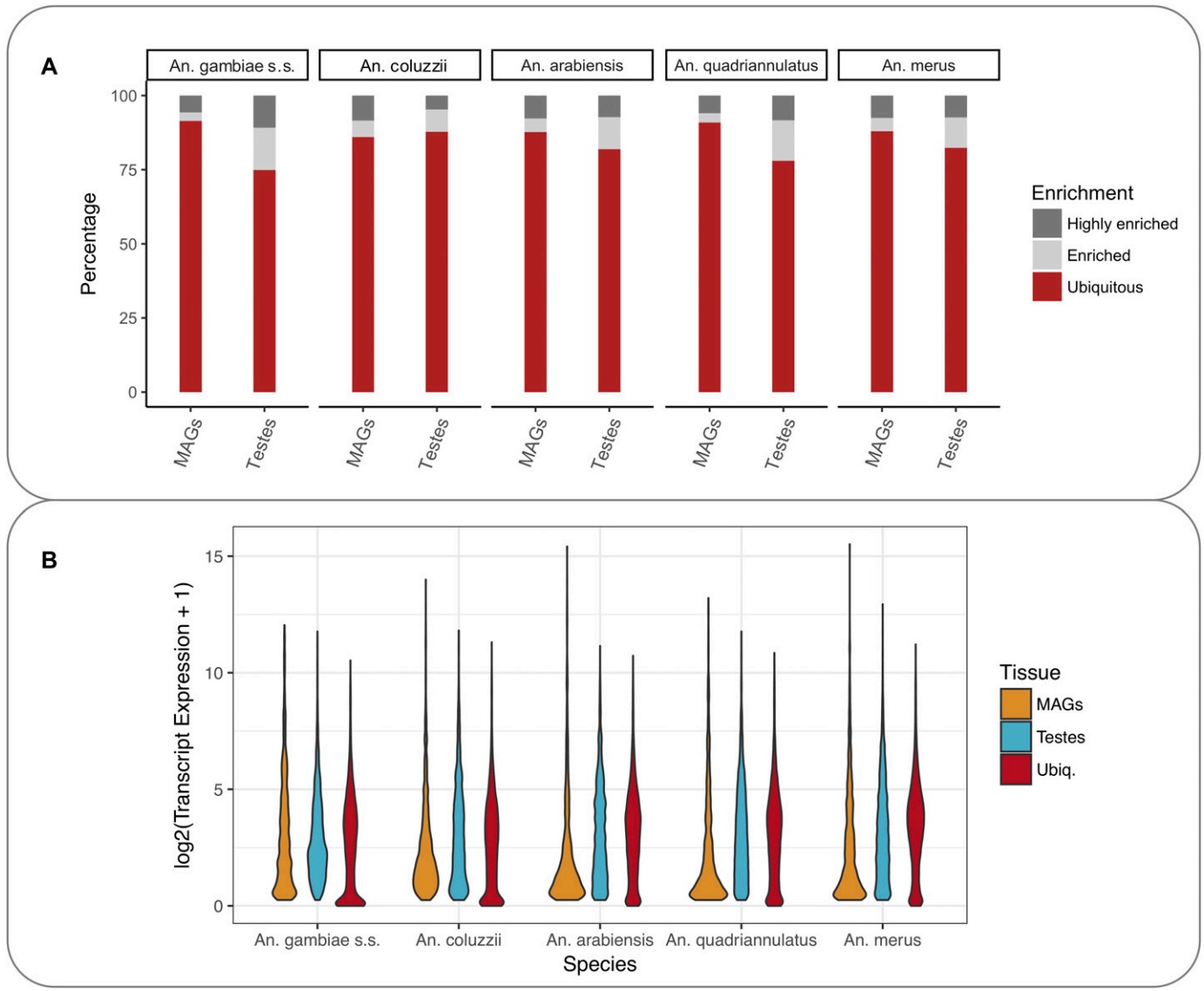

**Figure 1. MAGs and testes transcript expression levels in the *An. gambiae* complex.**
**(A)** Percentage of ubiquitous, enriched, and highly enriched transcripts in each species based on magnitude of transcript expression in that specific tissue versus the others. The expression levels of highly enriched, enriched, and ubiquitous genes in the different species were classified as highly enriched, enriched, and ubiquitous as defined in the text. **(B)** Distribution of enriched and ubiquitous transcript expression.

expressions in *An. gambiae*, 39.13% of *An. coluzzii*, 44.48% of *An. arabiensis*, 43.92% of *An. quadriannulatus* and 45.71% of *An. merus*. Inter-species expression normalisation allowed a direct comparison of expression across all species. To facilitate reading, the main findings are briefly summarised here:

- Via principal component analysis (PCA) and the construction of expression distance matrices, it was possible to investigate the phylogeny of the *Anopheles* complex directly from the analysis of expression divergence. This phylogeny agreed with the known anopheline phylogeny based on DNA sequence.
- Cluster analysis on expression patterns across tissues and species allowed the identification of two main types of interesting clusters: one containing transcripts with tissue-dependent expressions conserved across lineages and the other containing transcripts

with lineage-dependent expression patterns conserved across tissues.
- Both types of clusters could be further discriminated based on magnitude of expression change.
- Each type of cluster was found associated with specific molecular functions, transcription regulatory networks, and protein–protein interaction networks.
- Correlations were found between the expression patterns represented within the clusters, and strength of positive/negative selection.

**Comparison of gene expression across species**

PCA of expression levels of the ortholog transcripts showed not only a strongest separation between tissues but also a clear

differentiation between species (Fig 2A). The phylogeny based on the expression distance matrices of each tissue also agreed with the known anopheline phylogeny: separating *An. gambiae* and *An. coluzzii* into one clade from the three other mosquito species (Fig 2B).

Both the PCA and the expression distance results suggest the existence of time-dependent divergence of expression across species so that differences are less pronounced in the recently diverged species. This is consistently seen in MAGs and testes.

### Clustering and analysis of transcript expression patterns

Multivariate k-means clustering and Euclidean distance measurements of expression values in triplicates of the 5,493 orthologs identified 15 groups of transcripts with similar patterns across tissues and species (Fig 3). Eight clusters (1, 4, 7, 8, 11, 12, 14, and 15; Fig 3A), encompassing a total of 4,103 genes, did not exhibit any clear tissue- or lineage-dependent gene expression patterns. Seven clusters, encompassing a total of 1,390 genes, showed either tissue-dependent expression patterns conserved across the species (2, 3, 10, and 13; Fig 3B) or lineage-dependent expression patterns (5, 6, and 9; Fig 3C). For membership of each cluster, see Table S2.

#### Clusters featuring tissue-dependent expression patterns

Transcripts in clusters 2 and 10 show high expression in MAGs and low in testes, with cluster 2 showing higher MAGs expression than cluster 10. Clusters 3 and 13 show the opposite pattern (high expression in testes and low expression in MAGs) with cluster 3 having higher levels of expression in testes (Fig 3B). Since simultaneously expressed transcripts are often involved in common functional pathways, the four clusters 2, 10, 3, and 13 were inspected for statistically significant differences in enrichment in functional

classification (gene ontology [GO] biological process, molecular function and cellular component, and Kyoto Encyclopedia of Genes and Genomes (KEGG) pathways) (Fig 3D, E and Table S3).

#### Cluster 2: transcripts most highly enriched in MAGs

Within this cluster were three important sets of genes known to be key regulators of mosquito reproduction: (a) Acps (Dottorini et al, 2007); (b) 20-hydroxyecdysone (20E) synthetic pathway genes (Pondeville et al, 2008, 2013); and (c) genes known to participate in the synthesis of the mating plug (Rogers et al, 2009).

(a) Several Acps were found, known to cluster in the 3R fertilising island and to be exclusively or abundantly expressed in the MAGs (Dottorini et al, 2007). These were as follows: AGAP012680, AGAP009360, AGAP009365, AGAP009367, AGAP009368, AGAP0-09369, AGAP009370, AGAP009371, AGAP009372, AGAP009373, and AGAP012706. Amongst these, AGAP012680 and AGP009353 are the homologs of *D. melanogaster* male-specific opa-containing gene (MSOPA) that has been shown to be transferred to the female during copulation and to reach different targets including the ovary and spermatheca (Ravi Ram et al, 2005). AGAP009370 AGAP012706, and AGAP009369 are the homologs of *D. mela-nogaster* Acp53Ea, whilst AGAP009367 is the homolog of *D. melanogaster* Acp26Ab. Acp53Ea and Acp26Ab are genes known to be associated with sperm defence, that is, the ability of an ejaculate in the female to resist displacement by a second ejaculate (Clark et al, 1995). AGAP009360 and AGAP009365 have *Drosophila* orthologs that are not known to be expressed in the MAGs in the fruit fly. AGAP009365 (plugin) is discussed below.

(b) Three genes AGAP002429, AGAP000284, and AGAP001039 are part of the seven genes known to participate in the synthesis of the male-transferred steroid hormone 20-hydroxyecdysone (20E), a key regulator of monandry and oviposition in *An.*

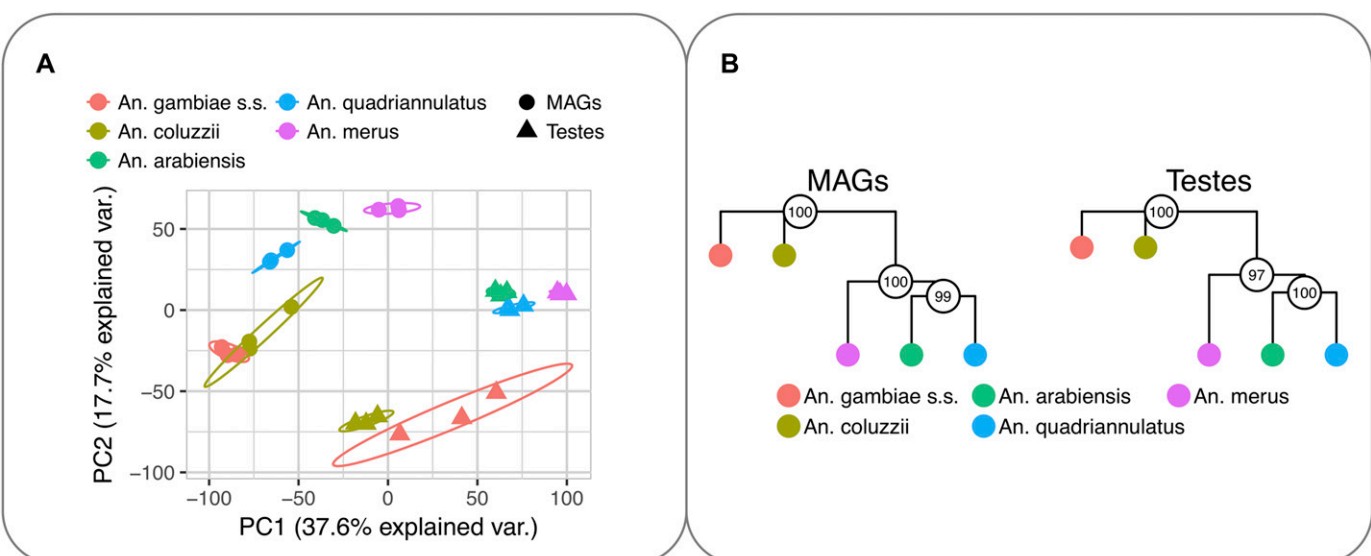

**Figure 2. Global patterns of gene expressions in the *Anopheline* species.**
**(A)** Factorial map of the PCA of the 5,493 orthologs expression levels. The percentage of the variance explained by the principal components is indicated in parentheses. Ellipses represent the 95% confidence intervals assuming normal distribution. **(B)** Transcript expression phylogenies (unrooted trees) based on the 5,493 orthologs. Neighbour-joining trees built on pairwise distance matrices (1—ρ, Spearman's correlation coefficient) are shown for MAGs (left) and testes (right). Bootstrap values (the 5,493 orthologs transcripts were randomly sampled with replacement 1,000 times) are shown by circles on the nodes.

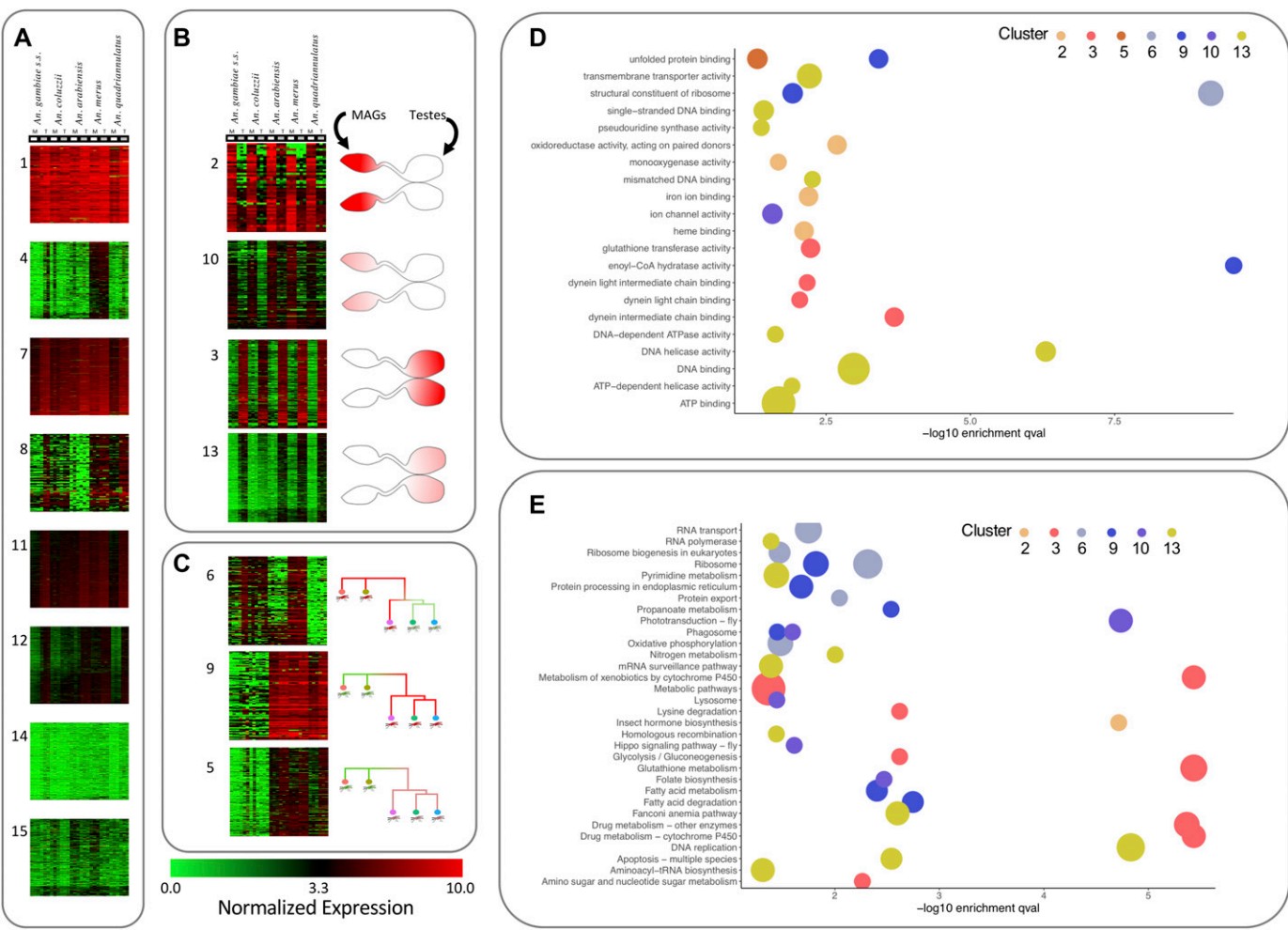

**Figure 3. Expression profiles of 5,493 transcripts grouped by k-means clustering.**
The 15 clusters have been categorised into three groups and are visualised as heat maps, and the colours in the heat maps indicate the level of expression. **(A)** The eight clusters (1, 4, 7, 8, 11, 12, 14, and 15) encompassing transcripts that did not exhibit any evident tissue- or lineage-dependent expression patterns. **(B)** The four clusters (2, 10, 3, and 13), encompassing genes showing tissue-dependent expression patterns. **(C)** The three clusters (6, 9, and 5) with lineage-dependent expression patterns. **(D)** The most highly enriched (–log$_{10}$ [*P*-value from hypergeometric test]) GO terms for the tissue- and lineage-dependent clusters. **(E)** Highly enriched KEGG pathways (–log$_{10}$ *P*-value from hypergeometric test) for the tissue- and lineage-dependent clusters. Circle colour represents cluster, and size is reflective of number of genes in cluster within a GO or KEGG category.

*gambiae* (Gabrieli et al, 2014) (Fig S1). These three genes encode cytochrome P450s and are part of the significantly enriched (hypergeometric test corrected *P* value < 0.05) KEGG pathway aga00981 "Insect hormone biosynthesis", important in ecdysone biosynthesis (Fig 3D).

(c) Two genes AGAP009368 and AGAP009099, encoding plugin and a transglutaminase, respectively, are known to be key genes for the synthesis of the mating plug (Rogers et al, 2009).

### Cluster 10: transcripts enriched in MAGs
The 147 members are enriched for transcripts, which encode the proteins involved in receptor, transporter, and ion channel activities (Table S3) and genes of known importance to reproduction (a) two Acps, and (b) two genes participating in the 20E synthesis.

(a) The Acps AGAP009353 and AGAP006582 are enriched in MAGs. AGAP006582 is a protease inhibitor located in the cluster of chromosome 2L (Dottorini et al, 2007). Despite AGAP006582

having a typical Acp structure and being located in the 2L cluster together with other experimentally validated Acps, this is the first confirmation of enriched expression in the MAGs.

(b) The 20E synthetic pathway genes AGAP000882 and AGAP005992 encode for a Glucose/ribitol dehydrogenase and a cytochrome P450, respectively (Fig S1).

In combination, these MAG-enriched clusters contained five of the seven known proteins for 20E synthesis (three found in cluster 2 and two found in cluster 10). One remaining gene AGAP001038 was found in cluster 14 (Fig S1). For the remaining gene in this pathway, AGAP004665's insufficient coverage meant that reliable orthologs could not be found across all species and it was therefore not part of our k-means analysis.

### Clusters 3 and 13: transcripts enriched in testes
Clusters 3 and 13 contain 216 and 499 transcripts, respectively. Cluster 3 contained an enrichment of genes involved in motor and

transport activity via dynein and microtubule-based movement and amino acid metabolism, supporting the fact that these genes are implicated with sperm movement. Cluster 13 contained genes enriched in ATP binding, and DNA replication activity again supporting the high replication activity of the testes.

### Lineage-dependent expression clusters

Three clusters (5, 6, and 9) showed a consistent expression between MAGs and testes but it differed between species (Fig 3C). Clusters 5 and 9 featured low expression levels in *An. gambiae* and *An. coluzzii* and high expression levels in *An. arabiensis, An. merus,* and *An. quadriannulatus*, cluster 9 had higher expression levels than cluster 5. Cluster 6 was more heterogeneous with lower expression in *An. arabiensis* and *An. quadriannulatus*, and high expression in *An. gambiae, An. coluzzii* and *An. merus*.

### Cluster 9: low expression in An. gambiae *and* An. coluzzii*, highest in* An. arabiensis*,* An. merus*, and* An. quadriannulatus

The 77 transcripts in cluster 9 encode proteins enriched in several metabolic pathways such as fatty acid elongation, degradation, and metabolism including genes AGAP005175, acetyl-CoA carboxylase/biotin carboxylase, and AGAP007784 an enoyl-CoA hydratase/long-chain 3-hydroxyacyl-CoA dehydrogenase (Table S3).

### Cluster 5: low expression in An. gambiae *and* An. coluzzii*, high in* An. arabiensis*,* An. merus*, and* An. quadriannulatus

Cluster 5 contained 273 transcripts enriched in heat shock protein binding and unfolded protein binding proteins (Table S3). When compared to VectorBase expression dataset, the 273 genes were not associated with expression in the male reproductive tract, for example, Magnusson et al, (2011) and Papa et al, (2017). The KEGG pathway analysis did not indicate any enrichment of functional pathways (Fig 3D).

### Cluster 6: low expression in An. arabiensis *and* An. quadriannulatus

The 130 transcripts in cluster 6 encode the proteins enriched in several metabolic pathways including metabolism of xenobiotics by cytochrome P450 (genes AGAP007374 and AGAP009192) shown to be down-regulated in response to 20E (Gabrieli et al, 2014) and proteins involved in targeting the ER (Table S3).

### Protein–protein interaction within and between clusters

Analysis of the male reproductive interactome showed that the lineage-dependent clusters (5, 6, and 9) are on average more highly connected (clustering coefficients 0.271, 0.305, and 0.397) compared with the tissue-dependent clusters (2, 10, 3, and 13), which generally showed lower coefficients (0.274, 0.169, 0.214, and 0.286). All but cluster 9 show lower connectivity than the complete *An. gambiae* protein–protein network (clustering coefficient 0.370). The average number of neighbours per protein ranged between 1.56 (cluster 2) and 17.05 (cluster 13). In terms of pairwise cluster-to-cluster connectivity (median pairwise cluster–cluster connectivity determined as number of edges connecting members of different clusters as a ratio of their combined number of nodes), tissue-dependent clusters 2, 10, and 3 had a lower pairwise cluster-to-cluster connectivity (0.75, 2.01, and 2.42, respectively) compared to lineage variable clusters 5, 6,

and 9 (3.03, 2.63, and 2.96, respectively). This suggests that the transcripts in cluster 2 encode for proteins that tend to have fewer interactions with proteins of other defined clusters. For cluster 2, the highest intercluster connectivity (1.62) was with the MAGS highly variable genes (discussed below).

### Transcription regulatory networks

By relaxing the strict constraints on ortholog identification (see the Materials and Methods section), we identified 328 transcription factors (TFs) present in all five species and for which we could find the expression in the original set of 13,000 transcripts. Amongst these, 147 TFs were found in our 15 clusters.

The 328 TFs were clustered on expression pattern, following the same method adopted for analysis of the original transcripts. Most of the TFs did not exhibit any tissue- or lineage-dependent expression pattern, showing conserved expression levels across all species and tissues instead. However, one cluster of TFs had a MAG-enriched expression pattern with marked differences between tissues, most like the transcript cluster 2; another had a testis-enriched pattern-like cluster 3, and a third one had a lineage-dependent pattern (low expressions in *An. gambiae* and *An. coluzzii* and high expressions in *An. arabiensis*, *An. merus*, and *An. quadriannulatus*)-like cluster 9. Protein–protein interaction (PPI) networks were isolated to verify whether expression pattern similarity would correspond to the known interaction between the TFs and the proteins encoded by the original transcripts (Fig 4; for cluster membership of TFs, see Table S2).

### MAGs-enriched TF networks

The TF cluster contained AGAP010358 (Gooseberry-neuro [*Gsbn*]), AGAP010359 (Paired [*Prd*]), AGAP008232 (Hamlet; no ortholog in *Drosophila*) and AGAP009699 (Growth factor 1; ortholog of *Drosophila melanogaster Senseless-2*). The PPI network identified seven first-degree interactions and five second–degree interactions with the proteins in cluster 2 (Fig 4A). Analysis of the seven first-degree interactions showed significant enrichment (hypergeometric test–corrected $P$ value < 0.05) in KEGG pathway aga00981 "Insect hormone biosynthesis" (genes AGAP000284, AGAP001039, and AGAP002429). The two TFs AGAP008232 and AGAP009699 interacted directly with AGAP001039, AGAP000284, and AGAP002429 (encoding enzymes in the 20 E pathway). This TF also showed a relatively high number of interactions with the second MAGs-enriched expression cluster 10 (17 primary [direct first degree interaction with TF] and 33 secondary second-degree interactions). Whilst the number of interactions is higher with cluster 10, as a percentage of the number of proteins in the cluster, this is slightly lower than for cluster 2 (19% and 14% of total proteins in the cluster are a primary interactor with the TFs for clusters 2 and 10, respectively).

### Testes-enriched TF networks

The PPI network indicated 27 first-degree interactions and 51 second–degree interactions with proteins in cluster 3 (Fig 4B). For cluster 3, the analysis of the first-degree interactors showed significant enrichment in gene ontology terms for cilium movement and microtubule-based movement, indicative of sperm activity/motility. For cluster 13, analysis of the first-degree interactors showed no enrichment of pathways and gene ontologies associated

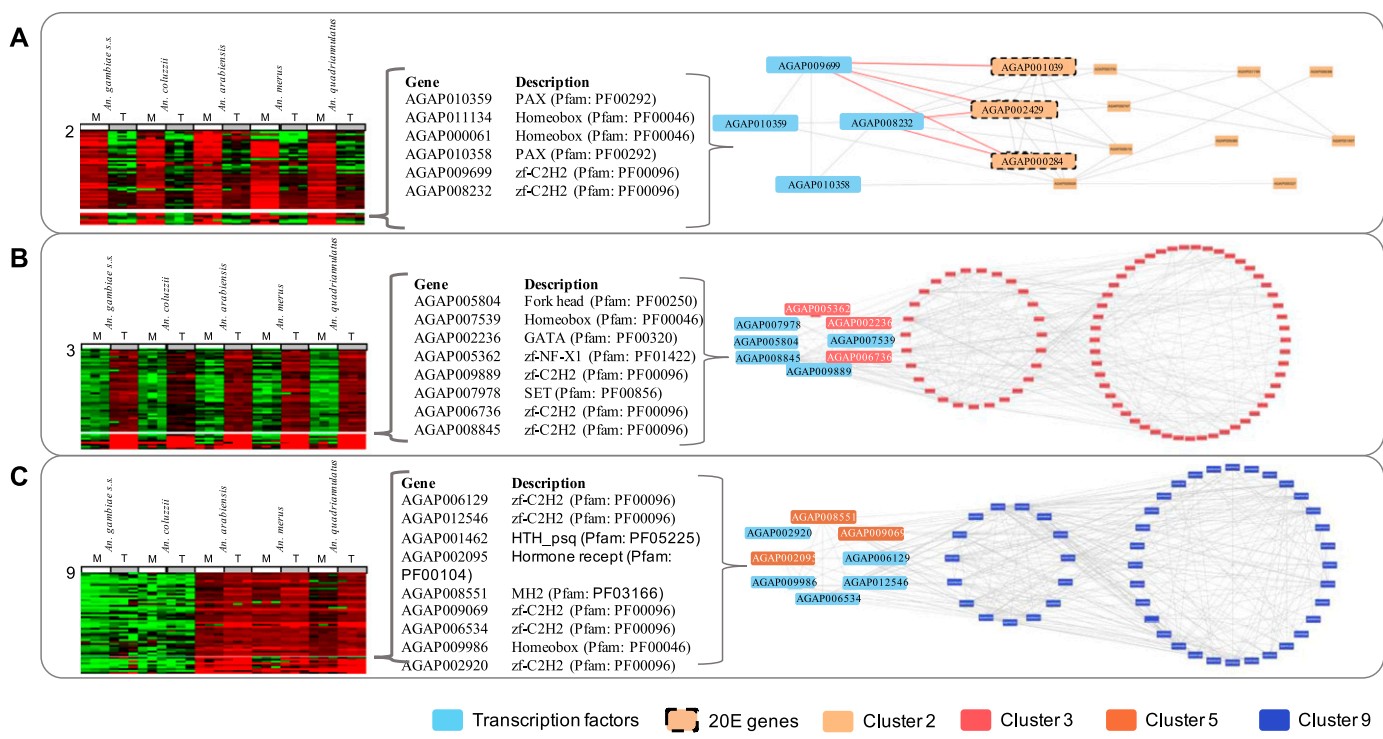

**Figure 4. TFs and predicted regulatory networks for the tissue- and lineage-dependent clusters 2, 3, and 9.**
Each panel shows the expression of transcripts and the TFs with most similar expression pattern. Names and descriptions of TFs in each cluster are shown (middle) and known protein–protein interactions of TFs with proteins in corresponding clusters are shown (right), TFs are in the dotted circles. **(A)** Cluster 2, MAGs enriched; **(B)** Cluster 3, testes enriched; and **(C)** Cluster 9, low expression in *An. gambiae* s.s. and *An. coluzzii*.

with DNA replication and transcription indicative of the high activity of the testes as a tissue. As before, the testes-enriched expression TF cluster showed a high proportion of primary interactors (17% and 24% of total proteins in the cluster are a primary interactor with the TFs for clusters 3 and 13, respectively).

### Lineage-dependent TF networks

The PPI network identified 15 first-degree interactions and 32 second–degree interactions with proteins in cluster 9 (Fig 4C). Three KEGG pathways: autophagy, endocytosis, and the spliceosome, were enriched in the primary interactors. Thirty-five first-degree interactions and 94 second-degree interactions are known for proteins in cluster 5. Again, these show a relatively high enrichment in the primary interactions (22% and 15% of total proteins in the cluster are a primary interactor with the TFs for clusters 9 and 5, respectively).

### Transcripts with highly variable expression across species

We analysed the 5,493 1:1 orthologs to identify the transcripts with the highest variation of expression across species. A total of 235 transcripts (4.2% of the total number) were identified in the MAGs (Fig 5A), and 287 (5.2%) were identified in the testes (Fig 5B) as highly variable, according to the procedure described in the Materials and Methods section. Both sets were inspected for functional enrichment (biological process, molecular function, and cellular component). For the MAGs, the most highly enriched functional terms were related to protein folding, cytochrome-c oxidase activity, and enoyl-

CoA hydratase. Amongst the genes found to be highly variable in the MAGs, we found five Acps (AGAP009370, AGAP012706, AGAP009369, AGAP009367, and AGAP009373). The first four had been classified as highly variable because of poor or no expression in the MAGs of *An. merus* whilst being significantly expressed in the other species. On the contrary, AGAP009373 was found expressed in the MAGs of all species but had low expression in *An. coluzzii*. Another Acp found to be highly variable was AGAP006582, expressed in all species except for *An. gambiae and An. coluzzii*. This gene had been previously identified as encoded in the Acp 2L cluster (Dottorini et al, 2007); however, it has never been experimentally confirmed as expressed in the male reproductive tract. Amongst the highly variable transcripts, we also found 11 other previously identified MAG-specific and MAG-enriched genes (see Table S3).

Less information could be obtained for highly variable transcripts in the testes. Comparison between genes in clusters to the published data at VectorBase did not find any clear correspondence with other expression studies including those focusing on the male reproductive tract (Magnusson et al, 2011; Papa et al, 2017). Significant enrichment in KEGG pathways included biosynthesis of amino acids, glycolysis/gluconeogenesis, and the metabolism of xenobiotics by cytochrome P450 (see Table S3).

The analysis of subnetworks corresponding to the highly variable transcripts indicated the testes and MAGs set as highly connected, and similar to the complete *An. gambiae* protein–protein network (clustering coefficient MAGs = 0.38, testes = 0.38 and complete network = 0.37). Analysis of the connectivity between the highly

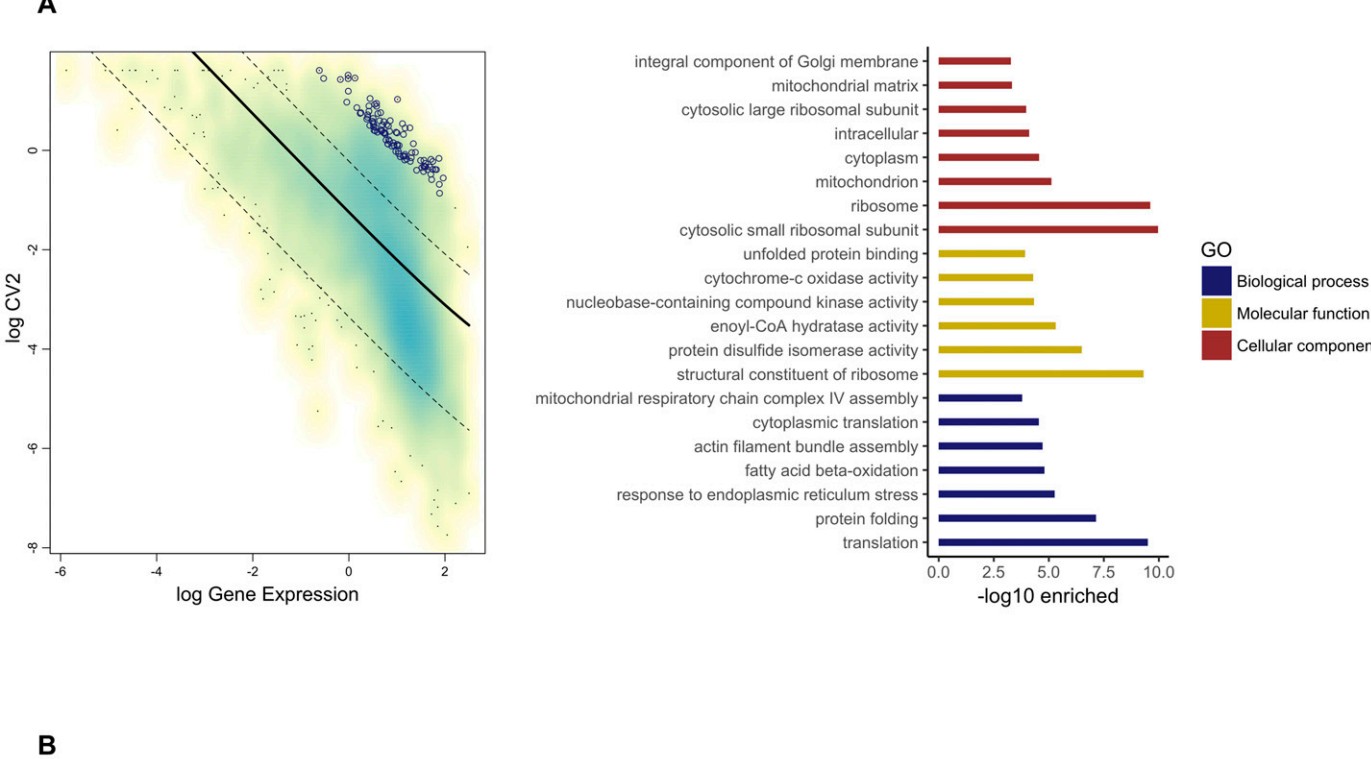

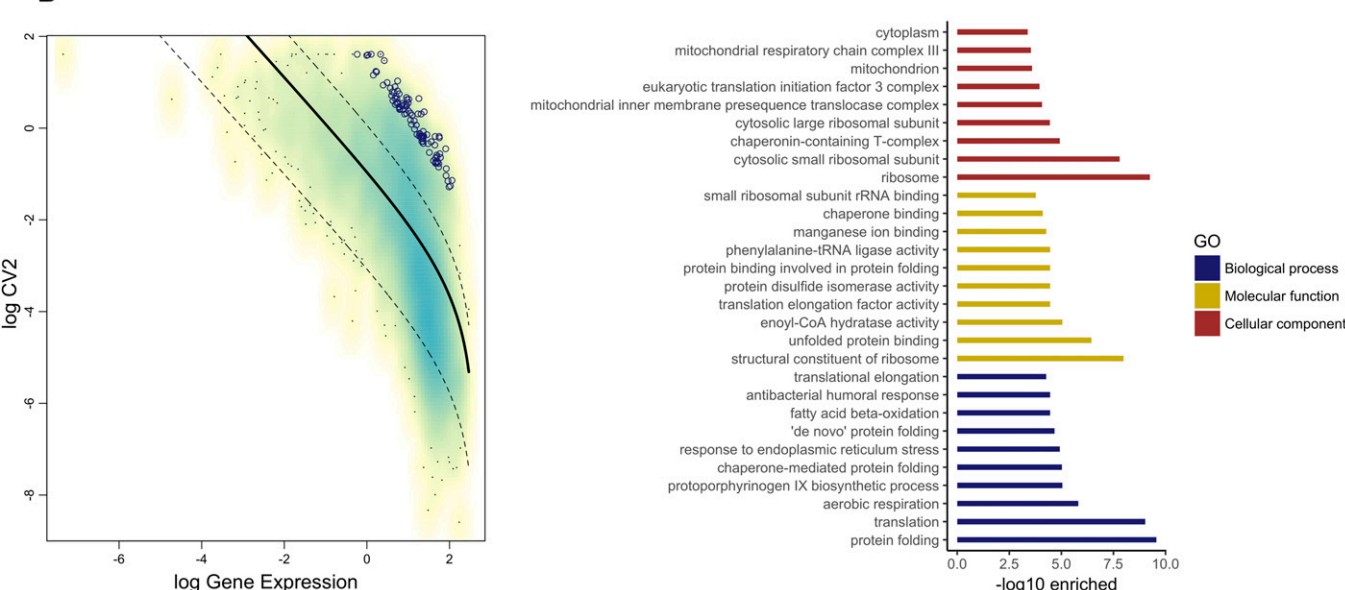

**Figure 5. Identification of highly variable transcripts.**
**(A)** Highly variable transcripts in the MAGs. **(B)** Highly variable transcripts in the testes. Transcripts identified as highly variable (ranked based on the significance of deviation from the fit) are shown as circles. The coefficient of variation of each transcript was plotted as a function of mean expression in a bi-logarithmic scale. The resulting scatter plot was fitted by linear regression, with confidence intervals at 95% confidence level. The most highly associated biological, functional, and cellular process GO terms with corresponding enrichment *P*-values are also shown.

variable sets and transcript clusters indicated higher connectivity of the testes and MAGs (3.61 and 4.64, respectively). For cluster 2 (high expression in MAGs, low expression in testes), which generally showed low connectivity to other clusters, the highest connectivity was observed with the highly variable MAGs set.

**Evolutionary rates and correlation with gene expression**

Analysis of the nonsynonymous to synonymous substitutions dN/dS ($\omega$) for the 5,493 1:1 ortholog transcripts showed that the majority of the orthologs exhibited strong purifying selection as seen in the

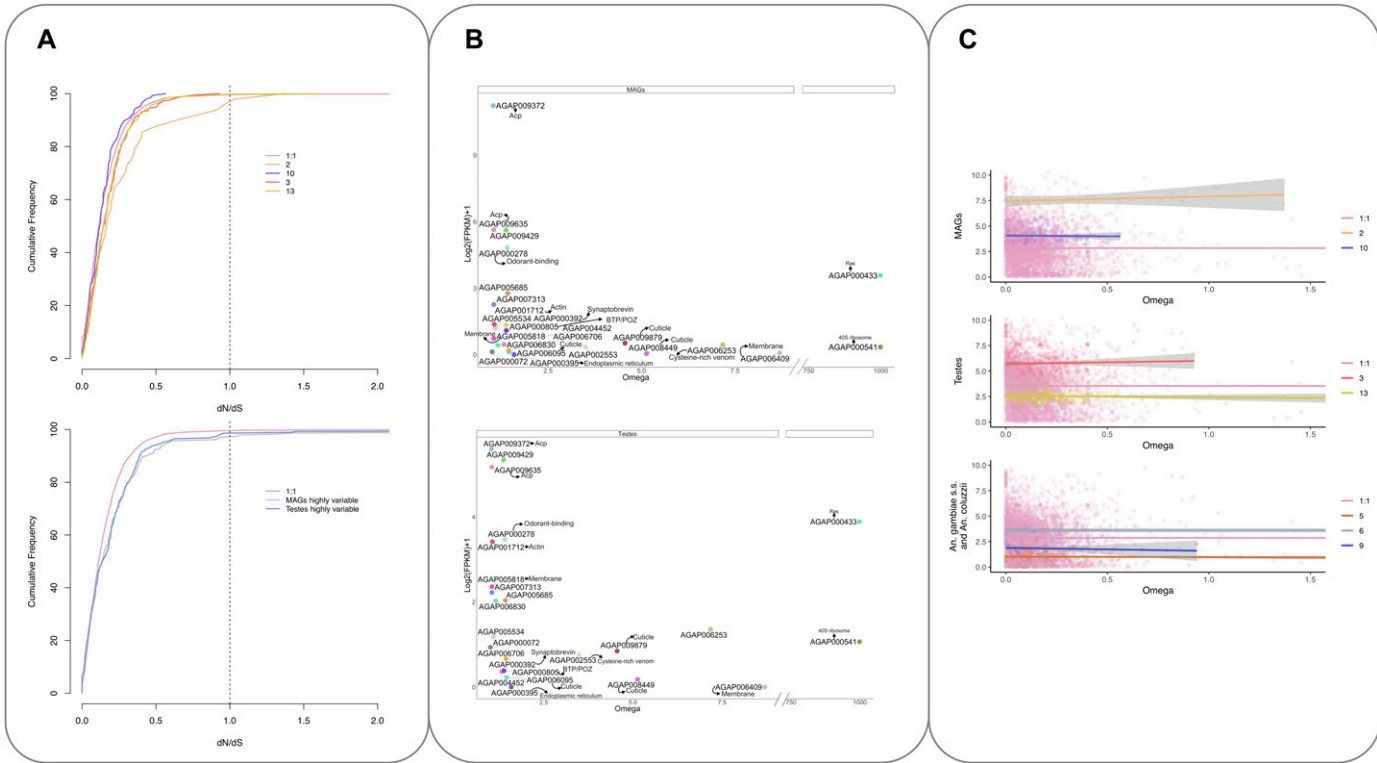

**Figure 6. Analysis of the nonsynonymous to synonymous substitutions dN/dS ($\omega$) for the 5,493 1:1 ortholog transcripts.**
**(A)** Cumulative frequency distribution of dN/dS ratios for the 5,493 orthologs (1:1) and tissue-dependent clusters (2, 10, 3, and 13) (top) and MAGs and testes highly variable genes (bottom). **(B)** Normalised expression in the MAGs (top) and testes (bottom) against $\omega$ values of the 24 transcripts showing evidence of dN/dS > 1 genes are coloured and labelled based on their available functional annotation. **(C)** Correlation between expression and evolutionary rate ($\omega$). The x-axis shows the ratio of nonsynonymous to synonymous substitutions (dN/dS or $\omega$). The y-axis shows the relative normalised expression (FPKM+1). MAGs-enriched clusters (top), testes-enriched clusters (middle), and lineage-dependent clusters (bottom) Clusters are compared to all 1:1 orthologs, which generally follow the trend that higher $\omega$ is associated with lower gene expression.

cumulative distribution of dN/dS (Fig 6A, upper panel). In contrast, cluster 2 (MAGs-enriched, highly expressed transcripts) shows a shift in the distribution to the right (Fig 6A, upper panel) reflective of a relaxation of selective constraints in these reproductive tissues, as previously seen in other species including mammals (Ramm et al, 2008). A similar shift was also observed in the MAGs and testes highly variable sets of transcripts (Fig 6A, lower panel). This shift is not seen in other tissue-enriched clusters.

Overall, 24 transcripts with $\omega$ > 1 were identified (Fig 6B and C and Table S4); 22 of these also showed statistically significant evidence of positive selection using paired models M1/2 and M7/8 and were predicted to have undergone rapid diversification since the divergence of the species complex (Table S4). No enrichment of function was seen, but four transcripts were annotated as encoding cuticular proteins (AGAP008449, AGAP006095, AGAP006830, and AGAP009879). The gene encoding a putative andropin-like AMP, AGAP009429 (cluster 2), found previously in the *An. gambiae* MAGs (Dottorini et al, 2007) and in the female after mating (Rogers et al, 2008) was also found to have an $\omega$ > 1. Also, in cluster 2 are two known Acps AGAP009372 and AGAP009365 (Dottorini et al, 2007), found to have $\omega$ > 1.

The correlation between evolutionary rate and gene expression was also investigated. There is generally thought to be a negative correlation between gene expression and evolutionary rate (Cherry,

2010); this was also seen for the majority of 1:1 orthologs analysed in this work (see Fig 6C). However, for the highly expressed tissue-enriched clusters (2 and 3) and lineage dependent cluster 6, there is a weak but positive correlation, suggesting a weak association between higher expression levels and increasingly relaxed purifying selection in these clusters (cluster 2, slope = 0.02 $R^2$ = 0.01; cluster 3, slope = 0.004, $R^2$ = 0.001; and cluster 6, slope = 0.01, $R^2$ = 0.006).

## Discussion

We analysed gene expression levels in two male reproductive tissues (MAGs and testes) of five closely related Anopheline species and investigated the relationships between changes in expression, DNA sequence divergence, transcriptional regulatory networks, and molecular functions. This analysis focused on closely related species, because gene expression changes, in contrast to DNA sequence variation, can vary on a large continuous scale and can evolve rapidly within a single lineage (Pollard et al, 2006). Therefore, a comparative assessment using distant lineages may not be able to account for multiple changes and hence may lead to imprecise species assignment.

Our results show that generally gene expression differences in the MAGs and testes of the five studied species are consistent with the known *Anopheles* phylogeny. Both multivariate analysis through

PCA and the construction of expression-based neighbour joining trees reflect the recent divergence of *An. gambiae* s.s. and *An. coluzzii*, and the separation of this clade from the other three species (*An. arabiensis*, *An. quadriannulatus*, and *An. merus*). These results suggest the existence of time-dependent divergence of expression in the MAGs and in the testes across species so that differences are less pronounced in the recently diverged species.

A detailed analysis of transcript expression across the five species and the two tissues based on k-means clustering revealed the existence of sets of transcripts with tissue-dependent expression patterns consistent across lineages, and sets of transcripts with lineage-dependent expression patterns. Cluster analysis also revealed that tissue- and lineage-dependent patterns can be further discriminated based on the level of expression. Importantly, predicted molecular functions and transcriptional regulatory networks are correlated not only with the type of pattern (e.g., tissue-dependent or lineage-dependent) but also with the level of expression. The most evident example of this is cluster 2, which contains MAGs-expressed transcripts with high expression levels, within which we found several of the most important regulators of reproduction in *Anopheles*: 11 Acps, 20-hydroxyecdysone (20E) synthetic pathway genes, and the only two known genes participating to the synthesis of the mating plug. Likewise in cluster 10, containing transcripts with similar MAG-enriched expression patterns, but comparatively lower expression levels, we again found the genes with the known reproductive functions, two Acps and two genes part of the 20E synthetic pathway.

Similar correlations were found for transcripts with lineage-dependent expression patterns. For example, clusters 9 and 5 populated by transcripts with low expression in the major malarial vectors *An. gambiae* and *An. coluzzii* and high expression in the other three vector species show enrichment of several metabolic pathways mostly related to signalling and defence, such as fatty acid, oxidative phosphorylation, and protein processing in the ER.

Fatty acids are compounds covering different biological roles from membrane structure to energy storage and have recently also been associated with reproductive behaviour in different insects. In bumblebees, fatty acids (linoleic acid) in the mating plug have key roles in eliciting refractoriness to remating (Baer et al, 2001). The sexual transfer of fatty acids from the male has been observed in several insects: in *Aedes albopictus*, a fatty acid synthase has been found to be transferred to the female after mating and possibly related to female nutrient usage after mating (Boes et al, 2014). American cockroaches, *Musca domestica* (Wakayama et al, 1986) and *Teleogryllus commodus* (Stanley-Samuelson and Loher, 1983; Worthington et al, 2015) are other insects for which fatty acids were found in the male reproductive tract. In the mouse, differences in fatty acids composition have been also associated with sperm mobility and competition (delBarco-Trillo et al, 2015). In *Anopheles*, the considerable divergence in plug phenotypes (coagulation) and content across species has been shown (Mitchell et al, 2015). The transcriptional differences observed in the fatty acid metabolism may represent signatures of evolutionary shifts between *An. gambiae* and *An. coluzzii* and other anopheline species, possibly inducing differences in reproductive traits including sperm or mating plug coagulation phenotypes; acting as female post-mating behaviour effectors (Neafsey et al, 2015); being required for the

production of sex pheromones; and controlling tissue-specific energy demand requirements. Enrichment in protein processing in ER was also detected. The ER is the major site for the translation and processing of secreted and membrane-bound proteins. The MAG is a major secretory tissue as it produces seminal fluid proteins that are secreted into the gland's lumen via the sec pathway. Several studies have shown that different *Anopheles* species do produce and transfer different amounts of MAG proteins. Clearly, genes playing a role in the translation and processing of the seminal proteins can impact the amount and the folding of the seminal proteins and hence male fertility.

To date, little is known about the transcriptional regulatory networks in the reproductive systems of the Anopheline species. To the authors knowledge, there has been only one previous paper by us (Dottorini et al, 2012) where such networks have been investigated and which identified the importance of the heath shock TF in the transcriptional regulatory networks of the MAGs of *An. gambiae*; the finding was experimentally verified in the same work.

Here, an original data mining method combining expression normalisation across species, clustering of transcripts, and TFs based on the patterns of expression levels, and protein–protein-interaction analysis, allowed identification of 23 putative TFs, which may form the basis for tissue and lineage-dependent expression patterns, in the MAGs and testes. PPI analysis also offers a tentative insight on the connectivity, composition, and configuration of these interacting subnetworks. Moreover, since our study involved comparison across five different species, we could also gain insight into which TFs are conserved across species. These are likely to represent a driving force in controlling species fitness.

It is important to note why the TFs coexpressed with the genes of a cluster were not necessarily found by PPI to interact with all the functionally relevant genes of that cluster. For example, the TFs found associated with cluster 2 were found interacting with genes responsible of the synthesis of 20E, but not with Acps, as it would be expected from the previous analyses. This is likely due to the lack of information currently available within the PPI databases, particularly for poorly understood proteins such as Acps. Whilst a complete understanding is not currently available, the identification of four Acps expressed in all species except *An. merus* and two expressed in all species except *An. gambiae* and *An. coluzzii* provides avenues for future investigation into speciation in these important vectors.

Some final interesting considerations can be made on the evolutionary constraints acting on the transcripts. As is expected, most transcripts exhibited strong purifying selection. However, the transcripts in cluster 2 (MAGs highly expressed transcripts) showed a relaxation of purifying selection, an indication of phenotypic plasticity that is reflected in the genome sequence. A similar trend was also observed for highly variable transcripts between species in both MAGs and testes. Twenty-two transcripts showed evidence of positive selection ($\omega > 1$) and were predicted to have undergone rapid diversification since the divergence of the species complex. Amongst these, two had been previously recognised as Acps.

Contrary to the negative correlation, which is often observed between gene expression levels and evolutionary rate, a slight tendency towards a positive correlation was observed for tissue-dependent

clusters (2 and 3) and for lineage-dependent cluster (6), which seems to indicate that some highly expressed genes are also more likely to be subjected to relaxed purifying, or positive selection. Although consistent, it is noted that this correlation is weak and should be confirmed and investigated across additional species.

In conclusion, we performed a large-scale analysis of the expression of transcripts and TFs involving five Anopheline species and two reproductive tissues (MAGs and testes). Thanks to interspecies normalisation and data mining, it was possible to quantitatively compare expression levels across species, leading to the identification of tissue- and lineage-dependent expression patterns. It was thus possible to observe that both expression patterns and expression levels correlate with molecular function, transcriptional regulatory network, and the interactome, suggesting an opportunity to investigate the evolutionary pathways of the mosquito through the analysis of the evolution of expression patterns. Our findings form a robust position to investigate these genes for mosquito research. Beyond *Anopheles*, our findings suggest promising venues of investigation in the study of gene expressions in rapidly evolving reproductive tissues.

# Materials and Methods

### Sample preparation

Mosquito stocks were obtained from the Malaria Research and Reference Reagent Resource Center (MR4) in Atlanta (https://www.beiresources.org/Catalog/BEIVectors/MRA-762.aspx). Specifically, *An. gambiae* s.s. (Savannah, MRA-762), *An. coluzzii* (Mopti, MRA-763), *An. arabiensis* (MRA-856), *An. merus* (MRA-1156), and *An. quadriannulatus* (MRA-1155).

Mosquito strains were maintained and reared in a climate chamber having stable temperature and relative humidity, that is, 27°C and 75%, with a 12:12 h light and dark regime. The populations of adult mosquitoes were maintained in plastic cages having the dimension of 18 × 18 × 18 cm (Acrilong) and fed ad libitum with 10% glucose plus 0.1% methylparaben as a preservative [6] from hanging sugar feeders. Mosquitoes were allowed to mate naturally within the cage, containing 50 males and twice as many females. To ensure the mating of all the male individuals, mosquitoes were left to mate for 4 d before dissection. MAGs and testes were dissected in PBS from 30 mated males at 5 d old. Tissues were isolated in RNAlater solution (QIAGEN) and stored at −80°C until RNA extraction. Total RNA was extracted from three biological replicates of 30 individual mosquitoes using the RNeasy Mini kit (QIAGEN) according to the manufacturer's instructions.

### RNA sequencing

Poly-A mRNA libraries were prepared in accordance with the Illumina TruSeq RNA sample preparation for Illumina Paired-End Indexed Sequencing. Samples were sequenced onto two lanes of an Illumina flowcell v3. The sample was then sequenced using the Illumina HiSeq1500, 2 × 100 bp paired end run.

### Gene expression quantification

RNA-seq reads from each sample were aligned to their corresponding reference genomes (*An gambiae* and *An. coluzzii* AgamP4.4; *An. arabiensis* AaraD1.4; *An. merus* AmerM2.2; and *An. quadriannulatus* AquaS1.4) using HISAT (Kim et al, 2015). Technical replicates from different lanes were merged using Samtools (Li et al, 2009) and then assembled in two steps to finally identify transcript level expression using StringTie (Pertea et al, 2015) (expressed as FPKM). The first assembly was performed using the corresponding reference genomes (Table S1) to identify the expression of all known transcripts in the species. A final merged assembly was generated from the de novo transcriptomes of all six samples of each species (three MAGs replicates and three testes replicates) using cuffmerge (Trapnell et al, 2012). For quantification, reads were aligned to the merged transcriptome. From the merged assembly, we used all novel transcripts in a species (not novel isoforms) and identified their orthologs in the *An. gambiae* reference genome with BLAST. Transcripts with a reciprocal best hit in *An. gambiae* and an E-value below $1 \times 10^{-10}$ were considered to have an ortholog and added to the transcript expression lists, identified by their *An. gambiae* s.s. ortholog.

### Inter-species expression normalisation

To normalise expression levels across species, we used a procedure presented by Brawand et al (2011), which calculates weighting factors for each species by finding the most stable transcripts from a list of known orthologs (here as defined by BioMart Version 869) and calculating normalisation factors for them (see Brawand et al [2011] for details). Using these factors, we adjusted our expression values across our five species.

The procedure is as follows: (a) a list of ortholog transcripts shared by the five species is identified (BioMart Version 869); (b) expression levels associated to the ortholog genes are retrieved from the original expression dataset. The orthologs missing expression data are removed; (c) since expression levels are available in triplicates, per transcript expression is preprocessed as log2 (mean(experimental triplicates) + 1); (d) within each sample, orthologs are ranked from the lowest (preprocessed) expression level to the highest. It follows that the same ortholog will potentially have different rank values in each sample; (e) the median and variance of rank values belonging to each ortholog across samples are computed. Orthologs with the highest median rank are those that are generally expressed the most across samples. Orthologs with the lowest rank variance are those whose expression levels remain the most constant across samples; (f) The orthologs whose expressions are generally very low across samples (low median) and the orthologs whose expression are generally very high across samples (high median) must be removed. This is done by ordering the orthologs on median rank value, in ascending order from the lowest, and then removing the first and last 25% (that is, only the orthologs within the IQR are kept); (g) the remaining orthologs are ordered according to variance of the expression rank across samples, from the lowest, in ascending order. This results in identification of transcripts whose level remains the most constant across samples. Of these, only the first tertile is kept for weight

computation; (h) these orthologs form a valid reference for the computation of the weighting factors and are referred to as "reference orthologs" (i) going back to the preprocessed expression (result of step (c)), the median of the expression of all the reference orthologs belonging to a sample is computed separately for each sample. The result is 10 median expression values, one per sample (five species × two tissues); (j) a "global mean" is computed as the arithmetic average of the 10 median expressions obtained at the previous step; (k) the weighting factor for each sample is computed as the expression median obtained at step (i), divided by the global mean obtained at step (j); (l) in the original dataset, all the expressions belonging to all the genes in all the samples can be multiplied by their associated weighting factor.

### Identification of one-to-one (1:1) orthologs

To identify the 5,493 1:1 orthologs, we compared *An. gambiae* s.s. transcripts with transcripts of the three species with different reference genomes using BLAST and gathered a list of transcripts with reciprocal best hits in all three other genomes. We further filtered for E-value $1 \times 10^{-10}$, percent identity 40%, and a maximum length difference between transcripts of 10% of the longest transcript length. Due to their known high sequence variation, orthology relationships of Acps were added manually.

### PCA and gene expression phylogeny

We performed PCA in R and plotted the first two principal components using ggbiplot (http://github.com/vqv/ggbiplot) with the ellipses representing the 95% confidence interval assuming normal distribution. For our phylogeny analysis, we calculated the distances between species using $1—\rho$, Spearman's-correlation as a distance measure. We then used the pvclust R library to calculate the bootstrapped hierarchical clusters and plotted the phylogeny trees using the Ape library (Paradis et al, 2004).

### Clustering

Cluster analyses were performed in MeV_4_8 (Multi Experiment viewer, mev.tm4.org) using k-means clustering with log2(FPKM+1) as expression values, Euclidean distance and k = 15 (number of clusters).

### Gene ontology and pathway analyses

The Bioconductor package GOstats (v2.42.0) (Falcon & Gentleman, 2007) was used to test for over-representation of GO terms using a hypergeometric test (hyperGTest). GO terms with a corrected *P* value < 0.05 were considered significantly enriched. Pathway analysis was performed using the Bioconductor package pathview (v1.16.5), which implements KEGG pathways. Significance enrichment level of KEGG pathways was identified by using false discovery rate < 0.05 and a corrected *P* value < 0.05.

### Protein–protein interaction

Protein–protein interaction data for *An. gambiae* were obtained from StringDB https://string-db.org/ using the criteria of "experimental,

database and textmining data". Redundant interactions were filtered resulting in a dataset of 1,534,953 pairwise interactions. Subsequent analysis and visualisation was conducted in Cytoscape version 3.6.1.

### Transcriptional regulatory networks

To search for TFs,: first, we scanned the 15 clusters obtained with the 1:1 ortholog dataset for the presence of annotated *An. gambiae* DNA-binding domain families as retrieved from the TF database. This led to the identification of 147 TFs. In addition, scanning all the entries from the TF database with 1:1 orthologs in *An. gambiae*, *An. arabiensis*, *An. quadriannulatus*, and *An. merus* using Biomart, and also measurable expression in our RNAseq dataset across the five species, resulted in a total 328 TFs. Clustering and PPI analyses were performed as described earlier.

### Highly variable transcripts

The coefficient of variation of each transcript was computed considering the expression values from all the species, plotted as a function of mean expression in a bi-logarithmic scale. The resulting scatter plot was fitted by linear regression, with confidence intervals at 95% confidence level. Data points (i.e., transcripts) were then ranked based on the significance of deviation from the fit and the most discrepant were selected as "highly variable". The top most-significantly-enriched GO terms were plotted against the negative $\log_{10}$ of the *P* value and coloured based to their functional category. *P* values were estimated based on a Chi-squared distribution and adjusted by FDR applying a cut-off of 0.05.

### Evolutionary rates and correlation with gene expression

Multiple sequence alignments of ortholog gene families were generated using the translated alignment function of Decipher (Wright, 2006) to generate a codon-based alignment. The ratio of nonsynonymous to synonymous sites (dN/dS or $\omega$) was calculated using model M0 of the codeML package from PAML (Yang, 2007).

## Data Access

The data discussed in this publication have been deposited in NCBI's Gene Expression Omnibus and are accessible through GEO Series accession number GSE117656.

## Supplementary Information

## Acknowledgments

The authors wish to acknowledge FP7 CAPACITIES/RESEARCH INFRASTRUCTURES Project INFRAVEC (228421) for funding the project: "ID0062—Genome expression profiling, DNA sequence divergence, and turnover analysis of *Anopheles*

species-restricted MAG genes". A Izquierdo was supported by an international PhD studentship from Consejo Nacional de Ciencia y Tecnologia (CONACYT) Mexico.

## Author Contributions

A Izquierdo: formal analysis and writing—original draft, review, and editing.
M Fahrenberger: data curation, formal analysis, and writing—review, and editing.
T Persampieri: data curation, formal analysis, and writing—original draft, review, and editing.
MQ Benedict: conceptualisation, formal analysis, supervision, funding acquisition, and writing—review and editing.
T Giles: data curation and formal analysis.
F Catteruccia: conceptualisation and writing—review and editing.
RD Emes: conceptualisation, data curation, formal analysis, methodology, and writing—original draft, review, and editing.
T Dottorini: conceptualisation, data curation, formal analysis, supervision, funding acquisition, methodology, project administration, and writing—original draft, review, and editing.

## Conflict of Interest Statement

The authors declare that they have no conflict of interest.

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
