## [Reviewer comments · Life Science Alliance]

Life Science Alliance

Evolution of gene expression levels in the male reproductive organs of *Anopheles* mosquitoes

Abril Izquierdo, Martin Fahrenberger, Tania Persampieri, Mark Benedict, Tom Giles, Flaminia Catteruccia, Flaminia Catteruccia, Richard Emes, Richard Emes, Tania Dottorini, and Tania Dottorini

Corresponding author(s): Tania Dottorini, University of Nottingham

Review Timeline:	Submission Date:	2018-09-12
	Editorial Decision:	2018-11-09
	Revision Received:	2018-11-24
	Editorial Decision:	2018-12-19
	Revision Received:	2018-12-21
	Accepted:	2018-12-21

Scientific Editor: Andrea Leibfried

Transaction Report:

DOI: 10.26508/lsa.201800191

November 9, 2018

Re: Life Science Alliance manuscript #LSA-2018-00191-T

Tania Dottorini

Dear Dr. Dottorini,

Thank you for submitting your manuscript entitled "Evolution of gene expression levels in the male reproductive organs of Anopheles mosquitoes" to Life Science Alliance. The manuscript was assessed by expert reviewers, whose comments are appended to this letter.

As you will see, the reviewers appreciate your work and make good suggestions on how to further strengthen it in a minor revision. Importantly, the work extends prior work, and this should be better reflected in the manuscript text (see also comments reviewer #2).

Thank you for this interesting contribution to Life Science Alliance. We are looking forward to receiving your revised manuscript.

Sincerely,

- A letter addressing the reviewers' comments point by point.
- An editable version of the final text (.DOC or .DOCX) is needed for copyediting (no PDFs).
- High-resolution figure, supplementary figure and video files uploaded as individual files: See our detailed guidelines for preparing your production-ready images, <http://life-science-alliance.org/authorguide>
- Summary blurb (enter in submission system): A short text summarizing in a single sentence the study (max. 200 characters including spaces). This text is used in conjunction with the titles of papers, hence should be informative and complementary to the title and running title. It should describe the context and significance of the findings for a general readership; it should be written in the present tense and refer to the work in the third person. Author names should not be mentioned.

B. MANUSCRIPT ORGANIZATION AND FORMATTING:

Full guidelines are available on our Instructions for Authors page, <http://life-science-alliance.org/authorguide>

Reviewer #1 (Comments to the Authors (Required)):

Izquierdo et al. analyze expression evolution in two male reproductive organs (testis and male accessory glands) across five closely related Anopheline mosquitoes. They identify groups of transcripts with expression variation across tissues but conservation across lineages or differentiation across lineages but conservation across tissues and find that these genes have the greatest associations with reproductive function, transcriptional regulatory networks, protein-protein sub-networks, and evolutionary rate. Overall, this is a nice, if rather descriptive analysis, divergence and conservation in expression in two tissues types across the Anopheles phylogeny and the methods are sound and conclusions are well supported. The authors very nicely apply the method of Brawand et al. 2011 to define different gene clusters, which in my mind is the main strength of the paper.

Other comments:

1. What do the authors mean by "adaptive evolutionary drift" on page 3? Is this a typo?
2. I was surprised that the introduction did not cite many papers with comparative analyses of gene expression across phylogenies or analyses of expression in reproductive tissues. These would provide important context for the paper. A few examples:
Rifkin, S.A. et al. (2003) Evolution of gene expression in the *Drosophila melanogaster* subgroup. *Nat. Genet.* 33, 138-144
Lemos, B. et al. (2005) Rates of divergence in gene expression profiles of primates, mice, and flies: stabilizing selection and variability among functional categories. *Evolution* 59, 126-137
Haerty, W. and Singh, R.S. (2006) Gene regulation divergence is a major contributor to the evolution of Dobzhansky-Muller incompatibilities between species of *Drosophila*. *Mol. Biol. Evol.* 23, 1707-1714
Moehring, A.J. et al. (2007) Genome-wide patterns of expression in *Drosophila* pure species and hybrid males. II. Examination of multiple-species hybridizations, platforms, and life cycle stages. *Mol Biol Evol.* 24, 137-145
Sundararajan, V. and Civetta, A. (2011) Male sex interspecies divergence and down regulation of expression of spermatogenesis genes in *Drosophila* sterile hybrids. *J. Mol. Evol.* 72, 80-89
3. What proportion of genes identified in any one species have orthologs in the other species and could be included in the analysis? I may have missed this point.

Reviewer #2 (Comments to the Authors (Required)):

This study states that its aim is to examine the evolution of gene expression in male organs of mosquitoes. Gene expression evolution is an important topic that deserves more attention in the literature, so I was glad to see a study on this issue. The study appears to assess evolution of gene expression in male reproductive organs of five species of *Anopheles*. However, the authors appear to downplay prior work by Papa et al. 2017, which has already done similar work in this genus of insects.

The authors briefly state in the introduction "A recent multi-species comparison investigated expression differences between sexes of four species (*An. gambiae*, *An. arabiensis*, *An. minimus* and *An. albimanus*) although this did not take into account different reproductive tissues (Papa et al. 2017)". This appears not to be accurate to me. Papa et al. studied both male and female reproductive and non-reproductive tissues in *Anopheles* including some of the same species studied here. For instance, here is a quote from Figure 2 in Papa et al. Fig 2 title "expression ratios (male/female) of carcass and reproductive tissue male-biased (blue) and female-biased (red) among the four-species orthologs". Moreover, Papa et al. 2017 studied the exact same male reproductive organs "testis and accessory glands" as studied here. If the authors are stating here in this sentence that they separately studied the testis and the accessory glands, and thus this study is different than Papa et al., this seems a very weak and vaguely stated argument. So, in my opinion there appears to be a serious downplaying on the prior work on this topic.

If I set that important issue aside, in my opinion, it certainly would be fine to publish this paper in addition to Papa et al. as the present study has gone into depth on analysis of specific male reproductive genes, and networks etc. This could make it a reference paper for genes of interest for future researchers. In addition, the figures are well done and the authors have clearly put a lot of work into their analyses.

The comments below are aimed to help the authors improve the manuscript.

1. Do not downplay results of Papa et al. 2017 or the various and obvious overlaps with the present study.

2. I would suggest the authors change the title to something like "A survey of gene and gene networks involved in male reproduction across five species of Anopheles". I was expecting a full study of expression evolution, and this appears more of a survey of genes expressed in male organs really, with only minimal study of expression evolution itself. Using this type of approach would also help with making the any novelty of this paper to Papa et al. 2017 more clear.

3. Results: The various sections on clustering results could be interesting. However, as presented, it is quite detailed, and hard to follow. What is the main question the authors are addressing here? It is not clear. It seems to be going to circles as presented now, and looks more like a survey of genes rather than testing a hypothesis or advancing our understanding of gene expression evolution. If it is a survey of genes, and a descriptive paper, not testing a specific hypothesis on expression evolution, then state this clearly. See point 2.

4. The authors should highlight only their main results as any message is diluted in by the minute details. Put the minor results in supplemental. And summarize this analysis in a more brief and clear manner.

5. Introduction: The authors mention "adaptive evolutionary drift". What is this? Adaptation and genetic drift in one?

The absence of line numbers makes it more tedious for reviewers to write comments. If a revision is submitted please add line numbers

Life Science Alliance manuscript #LSA-2018-00191-T

Evolution of gene expression levels in the male reproductive organs of *Anopheles* mosquitoes

Response to reviewers

Editor

...the reviewers appreciate your work and make good suggestions on how to further strengthen it in a minor revision. Importantly, the work extends prior work, and this should be better reflected in the manuscript text (see also comments reviewer #2).

We thank both Editor and Reviewers for their useful comments which we have addressed below and in the revised manuscript. We have acknowledged the important work by Papa et al (implied by reviewer #2 as the work we would be extending see page 4 of revised manuscript) and highlighted where our work differs. We feel it is important to say that we do not believe we are extending prior work. The work by Papa et al. has a different specific objective, a different scope of exploration, and applies different investigation methods. We believe that our paper represents an alternative approach to study evolution of gene sequence and transcript expression both approaches should be considered complementary but independent to each other.

We have provided a detailed explanation of this in the revised manuscript and in the response to Reviewer # 2 below.

In addition, we have reduced the length of the abstract to comply with the formatting guidelines of the journal.

Responses to Reviewer #1

Izquierdo et al. analyse expression evolution in two male reproductive organs (testis and male accessory glands) across five closely related Anopheline mosquitoes. They identify groups of transcripts with expression variation across tissues but conservation across lineages or differentiation across lineages but conservation across tissues and find that these genes have the greatest associations with reproductive function, transcriptional regulatory networks, protein-protein sub-networks, and evolutionary rate. Overall, this is a nice, if rather descriptive analysis, divergence and conservation in expression in two tissues types across the *Anopheles* phylogeny and the methods are sound and conclusions are well supported. The authors very nicely apply the method of Brawand et al. 2011 to define different gene clusters, which in my mind is the main strength of the paper.

Other comments:

1. What do the authors mean by "adaptive evolutionary drift" on page 3? Is this a typo?

That was improper use of terminology. We have rephrased the sentence.

2. I was surprised that the introduction did not cite many papers with comparative analyses of gene expression across phylogenies or analyses of expression in reproductive tissues. These would provide important context for the paper.

Thanks for the suggestion and for the list of publications. We have now added them where appropriate in the Introduction section.

3. What proportion of genes identified in any one species have orthologs in the other species and could be included in the analysis? I may have missed this point.

The information was indeed missing. The percentages of ortholog transcripts of each species have been now reported in the Results in a new paragraph entitled "Summary of the main results".

Responses to Reviewer #2

This study states that its aim is to examine the evolution of gene expression in male organs of mosquitoes. Gene expression evolution is an important topic that deserves more attention in the literature, so I was glad to see a study on this issue. The study appears to assess evolution of gene expression in male reproductive organs of five species of *Anopheles*. However, the authors appear to downplay prior work by Papa et al. 2017, which has already done similar work in this genus of insects.

The authors briefly state in the introduction "A recent multi-species comparison investigated expression differences between sexes of four species (*An. gambiae*, *An. arabiensis*, *An. minimus* and *An. albimanus*) although this did not take into account different reproductive tissues (Papa et al. 2017)". This appears not to be accurate to me. Papa et al. studied both male and female reproductive and non-reproductive tissues in *Anopheles* including some of the same species studied here. For instance, here is a quote from Figure 2 in Papa et al. Fig 2 title "expression ratios (male/female) of carcass and reproductive tissue male-biased (blue) and female-biased (red) among the four-species orthologs". Moreover, Papa et al. 2017 studied the exact same male reproductive organs "testis and accessory glands" as studied here. If the authors are stating here in this sentence that they separately studied the testis and the accessory glands, and thus this study is different than Papa et al., this seems a very weak and vaguely stated argument. So, in my opinion there appears to be a serious downplaying on the prior work on this topic.

If I set that important issue aside, in my opinion, it certainly would be fine to publish this paper in addition to Papa et al. as the present study has gone into depth on analysis of specific male reproductive genes, and networks etc. This could make it a reference paper for genes of interest for future researchers. In addition, the figures are well done and the authors have clearly put a lot of work into their analyses.

Our intention was not to downplay the work by Papa et al., but rather to point out that, apart for both papers comparing genes and transcripts between a number of *Anopheles* species, the two works are rather different in objectives, scope of exploration and methodological approach, and in our opinion should be seen as complementary ways to approach the investigation of the *Anopheles* phylogeny.

In detail:

The work by Papa et al. focuses on the exploration of general phenotypic and functional differences between sexes and how these are mapped across different species of *Anopheles*. Because of the focus on sex-biased genes, Papa et al. needed to consider both males and females, and, in order to maximise breadth of exploration, targeted the carcass on one side and an aggregate of multiple reproductive tissues on the other (MAGs and testis in males).

In our work we sacrifice breadth of scope to achieve greater exploration depth: 1) we are not searching for sex-biased genes, 2) we only focus on males, and 3) we only study male reproductive tissues. 4) Instead of aggregating reproductive tissues like in Papa et al., we keep MAGs and testes separate, which allows us to investigate not only how expressions are mapped across different species, but also how they map across different reproductive tissues.

There are another two fundamental differences that make ours and Papa's work stand apart. Papa et al. chose species that are spread across the entire *Anopheles* phylogeny, again to privilege breadth of exploration. On the contrary, we only focus on species that are closely related from the evolutionary standpoint, remaining narrowly confined within the *An. gambiae* complex. Our choice is mandated by the desire to perform a rigorous, quantitative comparison of expression: previous literature indicates that genes in reproductive tissues evolve very rapidly, thus the comparison of distant lineages could be dominated by significant DNA differences, leading to imprecise assignment of species and transcriptional changes (Pollard et al. 2006; Xu et al. 2018). On the contrary, by focusing on closely related species, we can rely on a more robust identification of correspondences, which allows a more reliable investigation of expression differences. Finally, we introduce inter-species expression normalisation (after Brawand et al, 2011), which allows for the first time a quantitative comparison of expression between orthologs in different anopheline species. This is highlighted by reviewer 1 as a particular strength of our manuscript. In Papa et al. it was only possible to perform quantitative comparison between tissues of the same species, whilst comparisons across species could only be performed from a qualitative standpoint.

In summary, we believe that, because of the explanations above, our paper should not be seen as an extension of the work by Papa et al. On the contrary, the two papers should be seen as different ways to approach what is clearly a very complicated problem, i.e. gaining a better understanding of the *Anopheles* phylogeny through the exploration of the evolution of sequences and expressions. Whilst sharing the common goal of contributing to improved understanding, the two papers differ in specific targets of investigation, scope of the exploration, and methods.

However, we recognise, as correctly pointed out by the reviewer, our depiction of the work by Papa et al. was probably too simplistic, which led to downplaying also the original contribution of our own work. We have now edited the Introduction, Results and Discussion sections of the manuscript to better clarify the differences with the work by Papa et al.

The comments below are aimed to help the authors improve the manuscript.

1. Do not downplay results of Papa et al. 2017 or the various and obvious overlaps with the present study.

(see previous response)

2. I would suggest the authors change the title to something like "A survey of gene and gene

networks involved in male reproduction across five species of Anopheles". I was expecting a full study of expression evolution, and this appears more of a survey of genes expressed in male organs really, with only minimal study of expression evolution itself. Using this type of approach would also help with making the any novelty of this paper to Papa et al. 2017 more clear.

Please do consider the answer to the previous question as important for setting the context to this answer as well. The reviewer's suggestion to change the title stems from a perceived, lesser contribution of our work to the study of expression evolution in anopheline, if compared to Papa et al. However, as stated above, we consider our work as an equally valid complement to the work by Papa et al., and not as a mere extension. In line with that, we disagree at depicting our paper as containing "only minimal study of expression evolution itself". As a matter of fact, despite us covering a narrower scope of species and tissues, we do believe our investigation goes more in-depth in the quantitative and comparative analysis of gene sequences and in particular of gene expressions inter-species.

In detail:

- 1) Similar to Papa et al., we have analysed the trend of the cumulative frequency distribution dN/dS ratios across species and tissues, however targeting different species and a different selection of tissues (considering an aggregate of reproductive tissues vs considering them separately has fundamental differences - see response to previous question).
- 2) We built for the first time a phylogeny entirely based on expression divergence. This could not be done in previous work because of lack of inter-species expression normalisation. This leads to an important and unprecedented result, because our phylogeny based on expression distance matrices also agreed with the known anopheline phylogeny, separating *An. gambiae* and *An. coluzzii* into one clade from the three other mosquito species (Figure 2B). Both the PCA and the expression distance results suggest the existence of time-dependent divergence of expression across species, so that differences are less pronounced in recently diverged species. This is consistently seen in MAGs and testes.
- 3) Because of being able to quantitatively compare expressions across species, we were also able to explore the correlation between evolutionary rate (Omega) and expression, across tissues and for the first time also across species. Again, this could not be done in previous work because of lack of inter-species expression normalisation.

3. Results: The various sections on clustering results could be interesting. However, as presented, it is quite detailed, and hard to follow. What is the main question the authors are addressing here? It is not clear. It seems to be going to circles as presented now, and looks more like a survey of genes rather than testing a hypothesis or advancing our understanding of gene expression evolution. If it is a survey of genes, and a descriptive paper, not testing a specific hypothesis on expression evolution, then state this clearly. See point 2.

The main results of the paper can be summarised as follows:

- 1) Via principal component analysis and the construction of expression distance matrices, it was possible for the first time to investigate the phylogeny of the Anopheles complex directly from the analysis of expression divergence. This phylogeny agreed with the known anopheline phylogeny based on DNA sequence.

- 2) Cluster analysis on expression patterns across tissues and species allowed the identification of two main types of interesting clusters: one containing transcripts with tissue-dependent expressions conserved across lineages, the other containing transcripts with lineage-dependent expression patterns conserved across tissues.
- 3) Both types of cluster could be further discriminated based on magnitude of expression change.
- 4) Each type of cluster was found associated to specific molecular functions, transcription regulatory networks and protein-protein interaction networks.
- 5) Correlations were found between the expression patterns represented within the clusters, and strength of positive/negative selection

Concerning the amount of reported results: in the manuscript we were faced with the challenging task of presenting results for more than five thousand orthologs, 15 identified clusters of transcripts each containing hundreds of transcripts, and multiple analyses conducted on such transcripts present in each cluster. The possibility of studying both sequence and expression across species for the first time dramatically expanded the amount of data to show, given the many analyses performed with a large array of different tools. This is why we decided to make the reading simpler by solely focus the presentation on the most interesting clusters and transcripts within each cluster.

Nevertheless, we understand that the amount of information provided may be a tad overwhelming, and – following the reviewer’s suggestion – we have edited the Results section of the manuscript, adding the paragraph “summary of main results” to summarise the main results and help the reader orienteering through the following, in-depth illustration of our findings.

4. The authors should highlight only their main results as any message is diluted in by the minute details. Put the minor results in supplemental. And summarize this analysis in a more brief and clear manner.

Most of the secondary results were already omitted from the original manuscript as not relevant (the uninteresting clusters were only mentioned, but not discussed in detail). A supplemental containing such information would clutter the paper with unnecessary information, and may distract the reader away from the relevant content. We hope that, through the clarifications introduced in the main text (see previous answer) we have now made the navigation of the reported result easier.

5. Introduction: The authors mention "adaptive evolutionary drift". What is this? Adaptation and genetic drift in one?

Improper use of terminology. We have rephrased the sentence.

December 19, 2018

RE: Life Science Alliance Manuscript #LSA-2018-00191-TR

Dr. Tania Dottorini
University of Nottingham
Sutton Bonington Campus
Sutton Bonington LE12 5RD
United Kingdom

Dear Dr. Dottorini,

Thank you for submitting your revised manuscript entitled "Evolution of gene expression levels in the male reproductive organs of Anopheles mosquitoes". As you will see, the reviewers appreciate the introduced changes, and only a few text changes as well as addressing some editorial points are still needed for acceptance. We would be thus happy to publish your paper in Life Science Alliance pending these final revisions:

- please address the comment of reviewer #2 by introducing text changes
- please add callouts to the text for Fig4A and Fig4C (currently you mention Fig4 as well as Fig4B)
- please link your ORCID iD to your profile in our submission system, you should have received an email with instructions on how to do so

A. FINAL FILES:

-- High-resolution figure, supplementary figure and video files uploaded as individual files: See our detailed guidelines for preparing your production-ready images, <http://life-science-alliance.org/authorguide>

B. MANUSCRIPT ORGANIZATION AND FORMATTING:

Full guidelines are available on our Instructions for Authors page, <http://life-science-alliance.org/authorguide>

Sincerely,

Reviewer #1 (Comments to the Authors (Required)):

No further comments.

Reviewer #2 (Comments to the Authors (Required)):

The authors did a nice job on the revision. I thought the original manuscript did not give enough (very minimal) credit to Papa et al., and it should have been cited throughout. I disagree with the authors on a couple points, particularly the overlap of the present work to Papa et al work, as I still see the present study as an extension of that prior work in many (not all) ways. But, that is fine, and we can have a difference of opinion on this. And I appreciate the value of the present results. Most importantly, the authors have credited the prior work by Papa et al. better in the revision.

I have only a couple points to add. I suggest the authors remove the words implying they are doing this type of analysis for the first time: "...but for the first time also across different species.". I believe Papa et al. did some analysis across species (in a different way than here), and journals usually prefer an author not to state they have done something for the first time. Again this is stated in the results "...for the first time to investigate" and again in the Discussion. So I suggest to remove throughout. Also, the words "both type of cluster" should state "both types of clusters".

The objectives, writing, and analyses are much clearer. The summary paragraph is quite nice. I believe the reviewer's job is just to make sure the work is solid, and it is put in context. And it is not to push the authors into making style changes etc they prefer not to make. So, I am not going to push any points on the title, or formatting etc.

Recommend to accept. Congratulations on a nice paper.

December 21, 2018

RE: Life Science Alliance Manuscript #LSA-2018-00191-TRR

Dr. Tania Dottorini
University of Nottingham
Sutton Bonington Campus
Sutton Bonington LE12 5RD
United Kingdom

Dear Dr. Dottorini,

Thank you for submitting your Research Article entitled "Evolution of gene expression levels in the male reproductive organs of *Anopheles* mosquitoes". It is a pleasure to let you know that your manuscript is now accepted for publication in Life Science Alliance. Congratulations on this interesting work.

DISTRIBUTION OF MATERIALS:

Again, congratulations on a very nice paper. I hope you found the review process to be constructive and are pleased with how the manuscript was handled editorially. We look forward to future exciting submissions from your lab.

Sincerely,
